# An 11 GHz Dual-Sided Self-Calibrating Dynamic Comparator in 28 nm CMOS

**Athanasios Ramkaj** [1,*] , **Maarten Strackx** [2], **Michiel Steyaert** [1] **and Filip Tavernier** [1]

1 ESAT-MICAS—KU Leuven, Kasteelpark Arenberg 10, B-3001 Leuven, Belgium;
Michiel.Steyaert@esat.kuleuven.be (M.S.); Filip.Tavernier@esat.kuleuven.be (F.T.)
2 Nokia, Bell Labs, Copernicuslaan 50, B-2018 Antwerp, Belgium; maarten.strackx@nokia-bell-labs.com
* Correspondence: Athanasios.Ramkaj@esat.kuleuven.be; Tel.: +32-(0)16-374-727

**Abstract:** This paper demonstrates a high-speed, low-noise dynamic comparator, employing self-calibration. The proposed dual-sided, fully-dynamic offset calibration is able to reduce the input-referred offset voltage by a factor of ten compared to the uncalibrated value without any speed or noise penalty and with less than 5% power overhead. Moreover, the implemented multi-stage topology significantly advances the state-of-the-art comparator performance, achieving the highest reported operating frequency, as well as the lowest delay slope and sensitivity to supply and common mode variations compared to existing works, with similar energy/comparison. This makes the proposed self-calibrating comparator an ideal candidate for high resolution (>10 b) multi-GHz Analog-to-Digital Converters (ADCs). The 28 nm bulk CMOS prototype measures an input-referred noise and calibrated offset of 0.82 mV and 0.99 mV, respectively clocked at 11 GHz, consuming only 0.89 mW from a 1 V supply, for an area of 0.00054 mm$^2$, including calibration.

**Keywords:** CMOS; dynamic comparator; offset calibration; high speed; low noise; low power; ADC

## 1. Introduction

Comparators are omnipresent building blocks in mixed-signal systems. Applications such as memories [1–3], data receivers [4–6], and Analog-to-Digital Converters (ADCs) [7–9] necessitate high speed, low noise/offset, yet power- and area-efficient designs. Their role in ADCs (Successive Approximation Register (SAR), flash, pipeline) (Figure 1) is of special importance, since they need to accurately translate small analog signals into digital information. Therefore, their noise, offset, and speed dictate the overall ADC performance.

Dynamic latch-type comparators [10–14] have become very attractive due to their fast regeneration time, enabled by strong positive feedback, and their zero static current consumption. Owing to their highly digital nature, these comparators are able to scale excellently into deep-submicron nodes. To maximize speed for minimal power, small transistor sizes with minimum parasitic loading on the critical nodes are preferred, but they come at the cost of a significantly increased offset [15,16].

One straightforward approach is to add amplification stages prior to the latch to suppress the offset voltage referred at the input [17]. This comes at the expense of increased power consumption due to the high gain and wide bandwidth requirements of these amplifiers. Alternatively, offset compensation schemes have been presented, in the form of adding digitally-controllable capacitors at the comparator outputs [18–20]. Further, several charge-pump implementations with extra logic and biasing voltages [21–23] have been proposed. However, all these approaches degrade the comparator and its calibration loop speed, increase design complexity and area, and limit robustness.

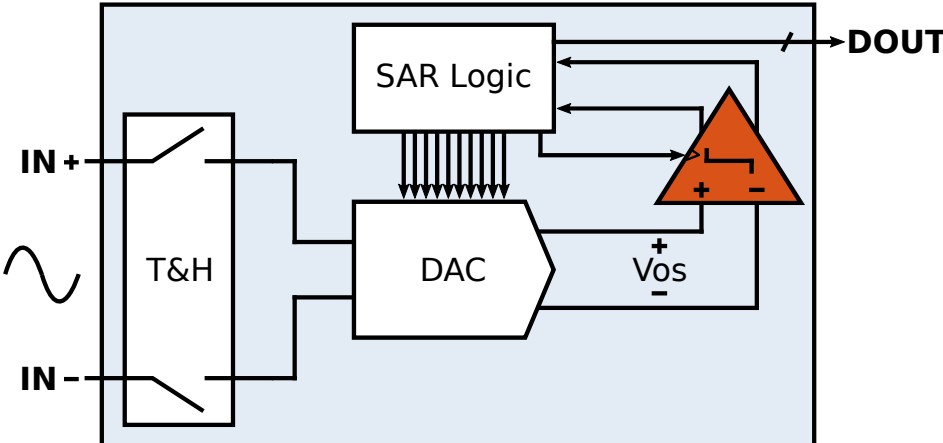

**Figure 1.** Top-level SAR ADC block diagram with its Track-and-Hold (T&H), DAC, comparator and SAR logic. A low-offset ($V_{OS}$), low-noise, and high-speed comparator determines the total performance and the accuracy on the output data (DOUT).

In this work, a dynamic self-calibration loop is proposed, able to reduce the comparator input-referred offset by a factor of ten. Its accuracy is limited only by the comparator noise, while its short critical path ensures no speed degradation. Finally, its compact size barely loads the comparator output. The loop's dual-sided implementation greatly enhances the calibration range, while its highly digital nature makes it readily scalable into deep-submicron nodes. Combined with a multi-stage, high-speed, low-noise dynamic comparator, >10 GHz operation is demonstrated with <1 mV input-referred noise (10 b accuracy), rendering it a perfect candidate for multi-GHz, high-resolution ADCs.

This paper is organized as follows. Section 2 describes the concept of the proposed dynamic comparator calibration. Section 3 discusses the circuit level implementation of the calibration and the comparator topology, supported by simulation data. Section 4 summarizes the measurement results along with a state-of-the-art comparison. Finally, Section 5 draws the conclusions of this work.

## 2. Dynamic Offset Calibration

The top-level architecture of the proposed offset calibration principle and its timing diagram are illustrated in Figure 2. The loop comprises a clocked comparator, two switched-capacitor calibration units (one for each side), and two offset compensating devices $M_{SP}$-$M_{SN}$. The calibration is performed simultaneously on both sides of the comparator (dual-sided), which maximizes the calibration range.

During calibration mode (CAL_EN is high), the common-mode voltage $V_{CM}$ is applied to both comparator inputs. For a positive offset voltage $V_{OS}$, the differential comparator output will be positive. The output sign is sensed by the two calibration units, which start subtracting charge from $C_{CALN}$ and adding charge to $C_{CALP}$, forcing nodes CALP/CALN to move in opposite directions to cancel this offset. When their difference reaches a certain value a$V_{OS}$, with a >1 depending on the size ratio of $M_{SP}/M_{SN}$ and the input transistors (see Section 3.2), the comparator differential output changes sign alternately. This means that the offset has been compensated, and the comparator now sees an input difference dictated only by noise. During conversion mode (CAL_EN is low), $C_{CALP}/C_{CALN}$ store the offset value, allowing the comparator to operate with canceled offset and decide correctly down to the noise level.

The comparator with the proposed calibration circuit can be easily incorporated in an ADC, where the already available periodic sampling clock can be used as CAL_EN, avoiding extra circuitry to generate that signal. Upon starting up the ADC, the calibration gradually corrects the comparator offset in multiple sampling cycles, by moving small packets of charge in each cycle until the required CALP/CALN difference is reached. This allows the offset calibration to run continuously in the

background, tracking supply noise, which can affect the input-referred offset, offering a true dynamic cancellation, while not interfering with the ADC operation.

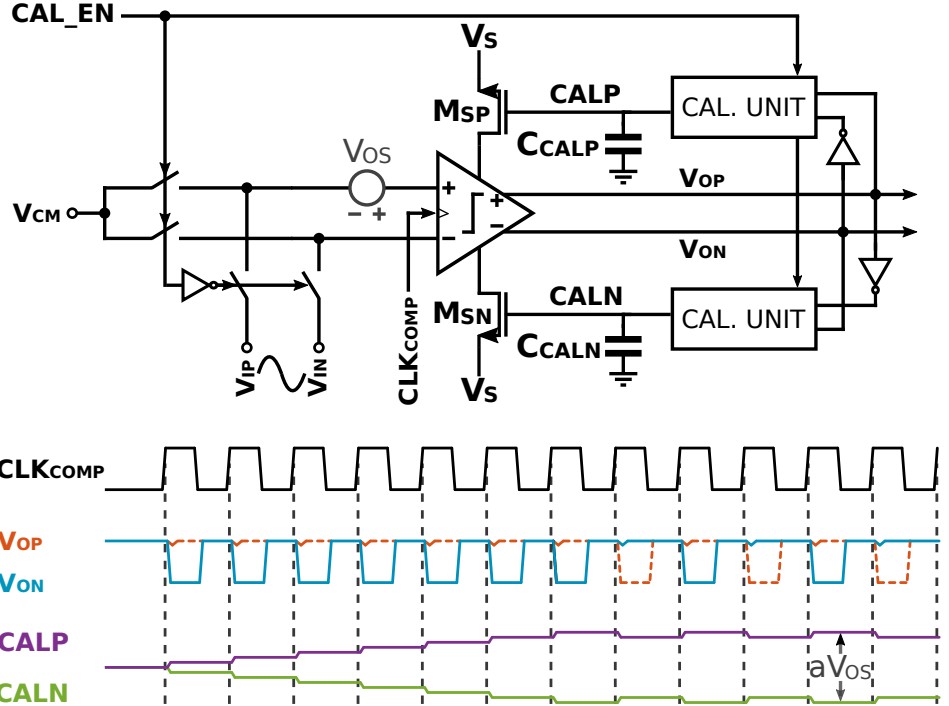

**Figure 2.** Top-level illustration of the proposed calibration (**top**) with its conceptual timing sequence (**bottom**).

## 3. Circuit Realization

### 3.1. Dynamic Self-Calibrating Loop

The single-ended version transistor-level implementation of the fully-differential self-calibration unit is shown in Figure 3. The unit consists of only eight switches plus one inverter. To eliminate the loading at the comparator output, all devices are minimum sized, while the fully-dynamic structure minimizes power overhead. Further, the delay between the comparator output and CALP/CALN is kept to a minimum of two transistors, such that the calibration loop does not impose a limitation on the total comparator speed. Unlike [21–23], there is no need for extra biasing circuitry to set the common-mode voltage on CALP/CALN. Here, it is gradually approaching the value set by proper sizing of the switches' on-resistance.

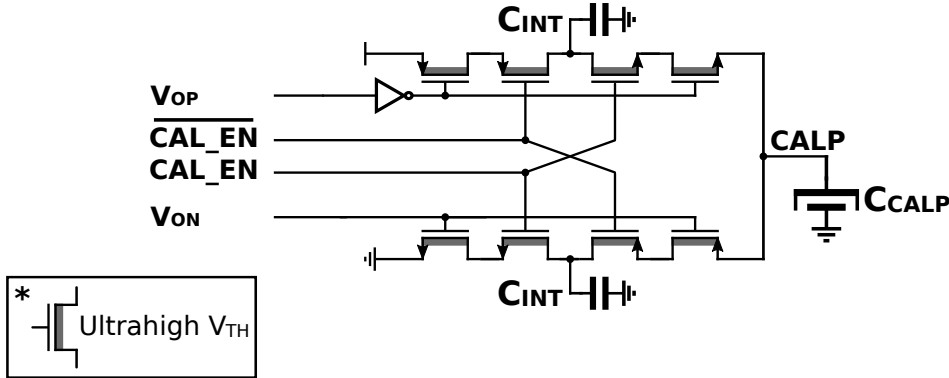

**Figure 3.** Transistor-level implementation of the self-calibration unit (single-ended shown for simplicity).

Charge is re-distributed between one of the internal capacitors $C_{INT}$ and $C_{CALP}$ and the amount of charge moved (calibration step) is controlled by the ratio of these capacitors and the time the calibration loop has available in each cycle, resulting in a wide compensation range. To reduce leakage on $C_{CALP}$, this capacitor has been constructed strictly as a Metal-Oxide-Metal (MOM) capacitor. Furthermore, ultrahigh $V_{TH}$ transistors are employed at the expense of more cycles required to compensate a certain offset value. In this way, the calibration step in each cycle is traded-off with the number of cycles. This is never a problem when testing an ADC, since there is always an allocated start-up time, after which useful data are collected and processed. The ultimate accuracy limitation of the proposed calibration technique is the comparator sensitivity to various conditions ($V_{DD}$ and/or $V_{CM}$). Therefore, a high-speed, low-noise, and low-input sensitivity comparator is needed to yield optimal results.

*3.2. Comparator Core*

The schematic of the comparator circuit where the proposed calibration is employed is shown in Figure 4. The comparator core incorporates a first amplification stage followed by a second amplifier/half latch and the final latch, in a fully-dynamic structure for low power operation [24]. The multi-stage configuration allows for a more orthogonal optimization of each stage for various trade-offs, which allows the comparator to simultaneously achieve both high speed and low noise.

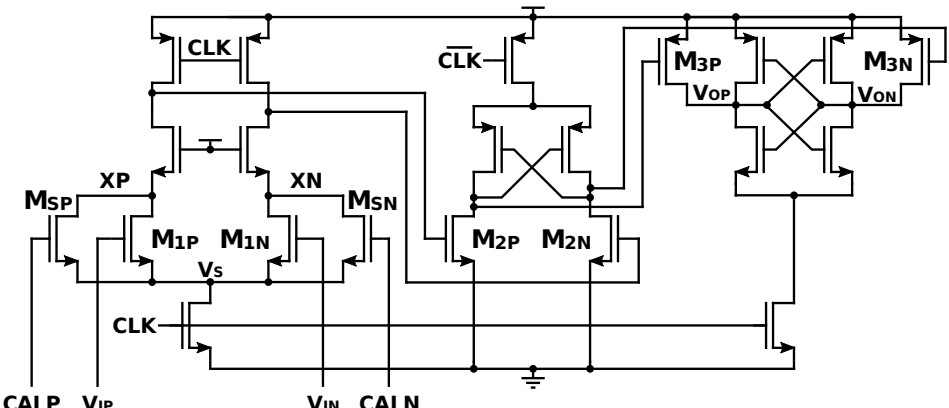

**Figure 4.** Comparator core with the extra offset compensating pair.

The required offset calibration pair $M_{SP}/M_{SN}$ is connected in parallel to the main input pair $M_{1P}$-$M_{1N}$. The differential gate voltage of the extra pair is varied in the opposite direction to that of the main pair, in order to reverse the offset. This additional input pair leaks the charge from XP/XN without integration, which deteriorates the comparator noise performance. Therefore, the dimensioning of the offset canceling pair is an important trade-off in terms of noise and calibration range. Larger transistors result in a larger calibration range, but also larger charge leakage/noise. In this design, the sizes of $M_{SP}/M_{SN}$ are chosen to be eight-times smaller than the main input pair $M_{1P}/M_{1N}$, to minimize the charge leakage, thus the noise degradation. This translates to a maximum of 125 mV offset compensation range (a$V_{OS}$ = 1 V in Section 2) for a common-mode voltage of 0.5 V, which is large enough to allow a low-power and high-speed comparator design.

The implemented self-calibrating comparator performance in terms of delay and noise has been characterized with extracted simulations and compared to the comparators from [10,11,25,26], scaled to 28 nm (Figure 5). For the comparator delay, the Overdrive Recovery Test (ORT) [27,28] has been used, while the noise has been characterized with both pss + pnoise and transient simulations. The operating conditions were $V_{DD}$ = 1.0 V and $V_{CM}$ = 0.5 V. The proposed design achieved more than 20% faster regeneration time for small inputs due to the increased gain in the signal path and showed a lower input dependency for a wide range of voltages compared to [10,11,25,26] (Figure 5a). To achieve similar regeneration times, the tail, as well as the latching transistors of the works in [10,11,25,26] had

to be upscaled, whose combined contribution increased the total input-referred noise by more than 15% with respect to this design, as shown in Figure 5b.

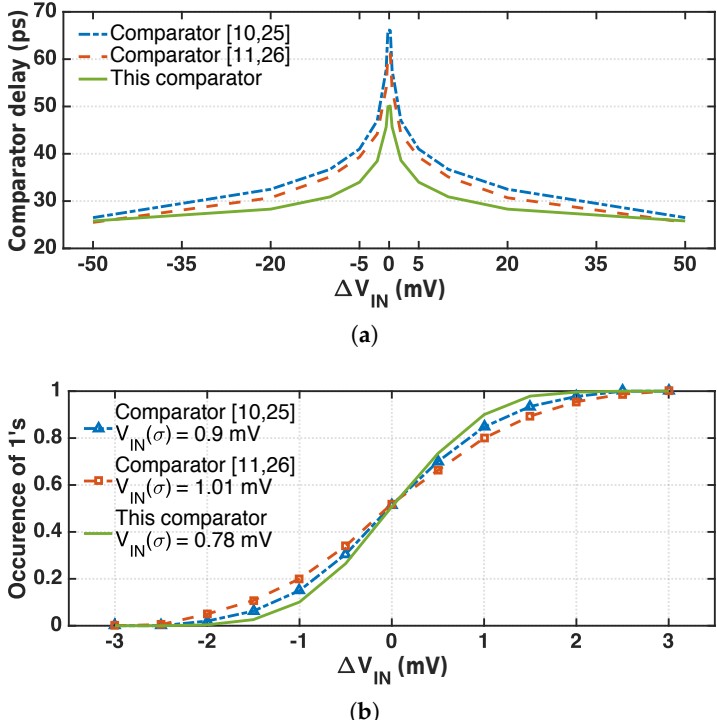

(a)

(b)

**Figure 5.** Simulated delay versus $\Delta V_{IN}$ for the same offset/noise (**a**) and cumulative noise distribution for similar delay (**b**) for [10,11,25,26] and the proposed design.

The offset for the three circuits with similar regeneration times has also been characterized through Monte Carlo simulations on 100 samples (Figure 6). A servo-loop has been used, which senses the comparator output and feeds back to the input the opposite offset value until the comparator goes into a metastable state. For the designed $V_{CM}$ of 0.5 V, the 1-$\sigma$ raw value for both [10,11,25,26] was larger than 11 mV, while it was 9.8 mV for the proposed design (Figure 6a). After enabling the proposed calibration, the offset was improved to 0.69 mV, set by the designed $C_{INT}/C_{CAL}$ ratio and noise, without compromising the rest of the specifications.

To show the effectiveness of the calibration for different common mode conditions, the comparator $V_{CM}$ was varied between 0.4 V (Figure 6b) and 0.6 V (Figure 6c). It is seen that the accuracy of the calibration loop remained functional for a wide range of common mode voltages, dictated only by the comparator noise.

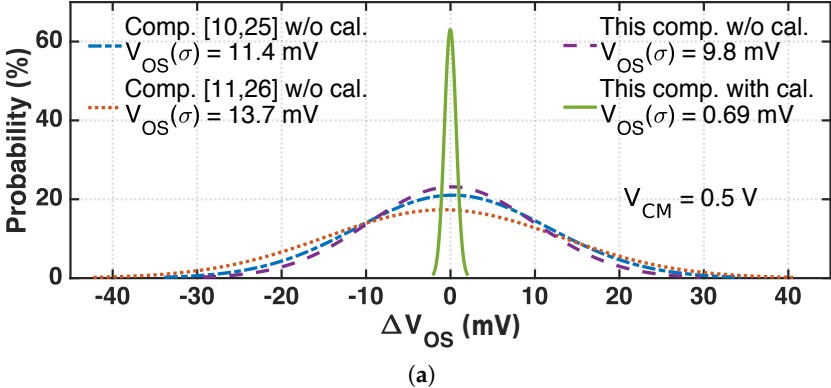

(a)

**Figure 6.** *Cont.*

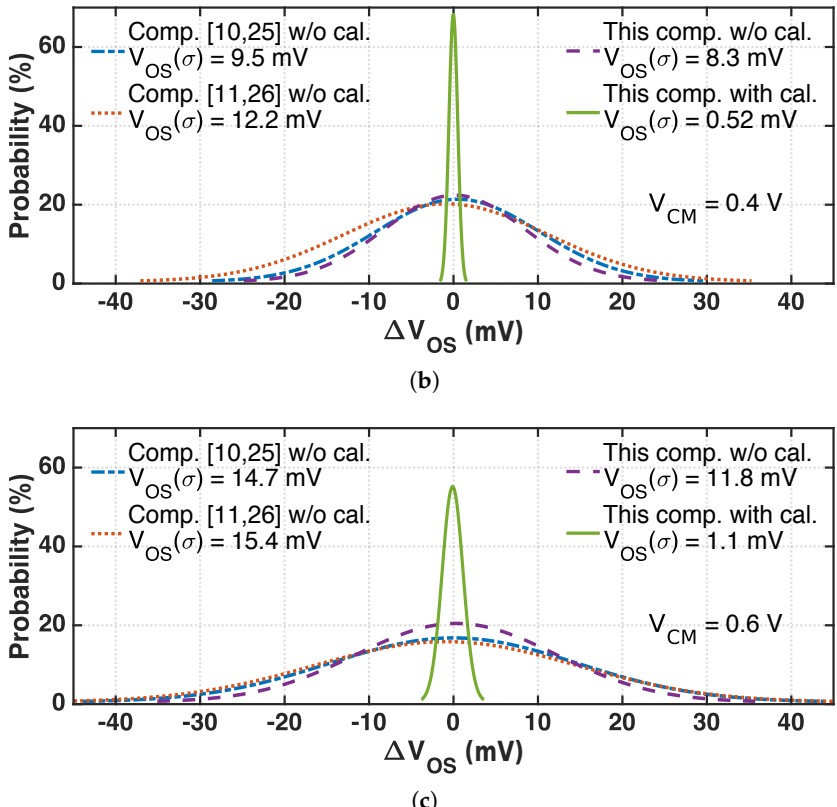

(b)

(c)

**Figure 6.** Simulated offset distribution for a $V_{CM}$ of 0.5 V (**a**), 0.4 V (**b**), and 0.6 V (**c**); for [10,11,25,26] and the proposed design.

## 4. Experimental Results

### 4.1. Measurement Setup

The measurement setup used to evaluate the comparator performance is shown in Figure 7. A low phase noise signal source (Agilent E8257D) was used to generate the up to 11 GHz sinusoidal clock signal. This signal was converted into to a square pulse through on-chip CML + CMOS circuitry. An identical signal source was employed to generate the comparator input signal. Both input and clock signals were converted into differential signals by two identical wideband hybrids and AC-coupled to the chip through custom-designed bias-tees and phase-matched cables. A dual-channel source-meter was used to bias the differential comparator input, for easier noise and offset extraction.

The signal generators were locked together and with a 63 GHz bandwidth scope (DSOZ634A), serving as a data analyzer, which captured the differential output at full speed. The captured data were then processed on a PC in MATLAB. First, the comparator noise was characterized by observing the data, while the raw offset was subtracted from the comparator by applying different DC voltages from the source-meter. After noise characterization, the calibration loop was enabled and the calibrated comparator offset, as well as speed were evaluated.

The required supply and bias voltages for the different chip domains were generated with dedicated low-noise Low-Dropout Regulators (LDOs) on a custom bias board and provided to the chip after sufficient low-pass filtering.

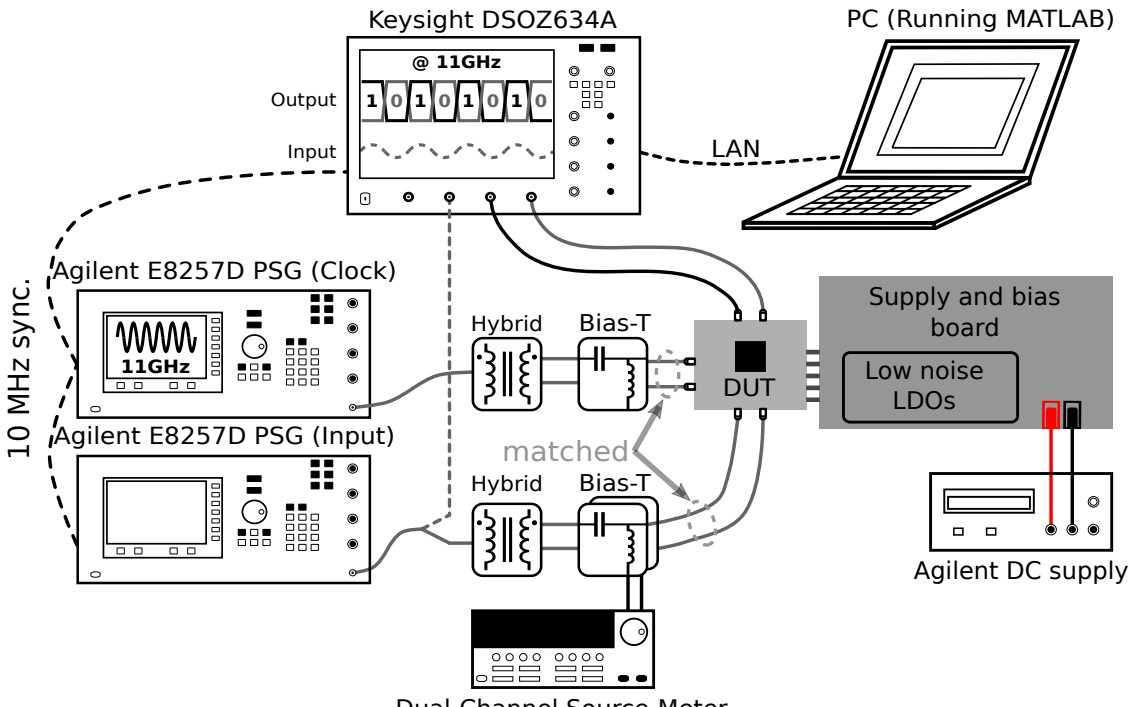

**Figure 7.** Measurement setup of the proposed self-calibrating comparator. LDO, Low-Dropout Regulator.

### 4.2. Measurement Results

The prototype self-calibrating comparator was realized in a single-poly ten-metal (1P10M) 28 nm bulk CMOS process and occupied an area of $35.5 \times 15.2\,\mu m^2$ (Figure 8). Most of the area was taken up by $C_{CALP}/C_{CALN}$, which is insignificant when used in an ADC. The measured power consumption of 0.89 mW at 1 V and 11 GHz clock frequency ($F_{CLK}$) partitions into 0.87 mW for the comparator core and only 0.02 mW for the calibration logic, less than 5% overhead.

Figure 9 illustrates the measured input noise versus $\Delta V_{IN}$ when varying $V_{DD}$ (top) and $V_{CM}$ (bottom), respectively, at 11 GHz. The noise was extracted by counting the percentage of positive decisions with increasing the differential input voltage, having first subtracted the comparator offset. The calibration loop was disabled for this measurement. The comparator measured a 1-$\sigma$ noise voltage of $0.82\,mV_{rms}$ for $V_{DD} = 1\,V$ and $V_{CM} = 0.5\,V$, which varied by only $0.13/-0.14\,mV$ when $V_{DD}$ changed from 0.9 V to 1.1 V and $-0.19/0.18\,mV$ when $V_{CM}$ changed from 0.4 V to 0.6 V.

The offset voltage of the comparator was measured across 15 chips operating at 11 GHz with $V_{CM} = 0.5\,V$ for the maximum compensation range (see Section 3.2), as shown in Figure 10. The offset was extracted by sweeping the input voltage of the comparator until the ratio of zeroes and ones was ∼50%. For the raw offset value, the calibration loop was disabled, while for the compensated value, the calibration was activated prior to collecting the data. Thanks to the proposed dual-sided calibration technique, the offset voltage was drastically reduced to 0.99 mV from the uncalibrated 10.3 mV (>10× improvement), without compromising the comparator speed, verifying its smooth integration in single- or multi-comparator ADCs. As expected, the calibration accuracy was ultimately limited by the comparator noise, which also matched nicely with the simulated results.

The maximum speed of the comparator for small inputs, close to the noise level, was characterized by observing the frequency above which increased differential input was required to preserve correct digital outputs. An eye diagram is shown in Figure 11 for an $F_{CLK}$ of 11 GHz and a coherent Nyquist sinusoidal input frequency of 5.46 GHz (= (8133/16,384) × 11 GHz), such that the comparator can capture any input voltage value over the full-scale range. With this setup, no metastability errors were detected for voltages outside the comparator noise levels, measured over one million time samples.

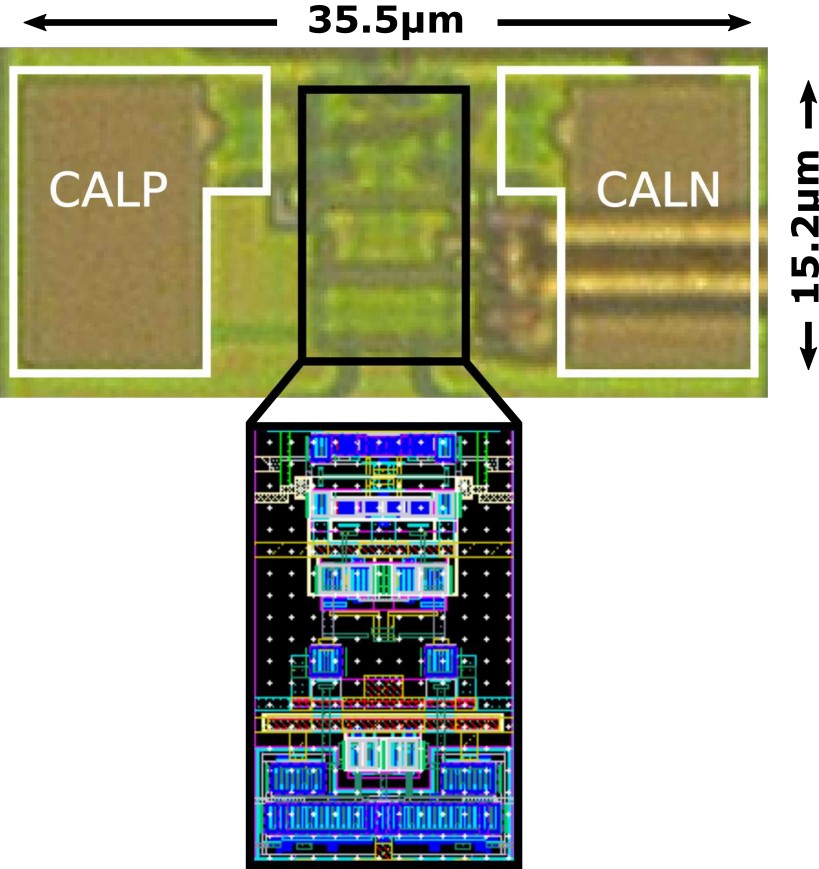

**Figure 8.** Die photo of the 28 nm self-calibrating comparator with a layout view of the comparator core.

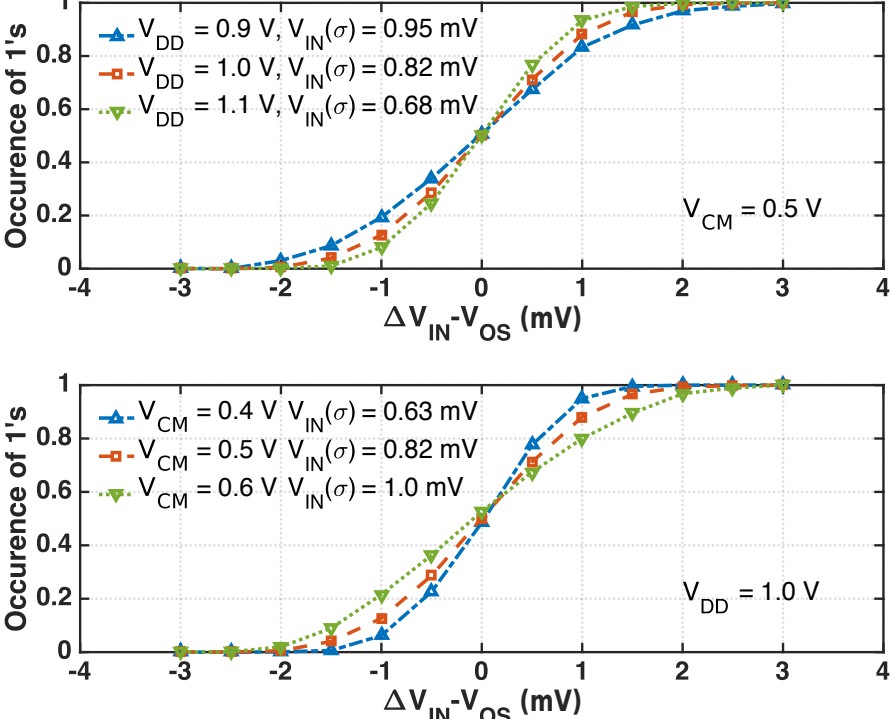

**Figure 9.** Measured cumulative noise distribution versus differential input at 11 GHz for varying $V_{DD}$ (**top**) and varying $V_{CM}$ (**bottom**).

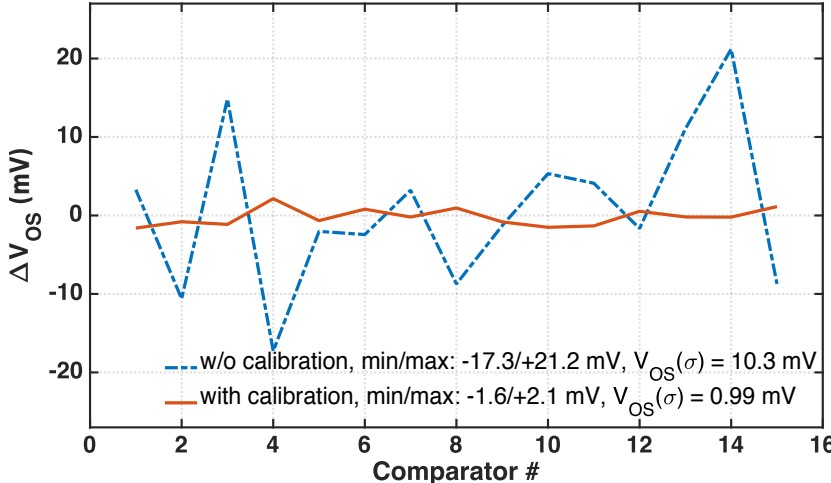

**Figure 10.** Measured raw and calibrated comparator offset voltage with the proposed calibration at 11 GHz.

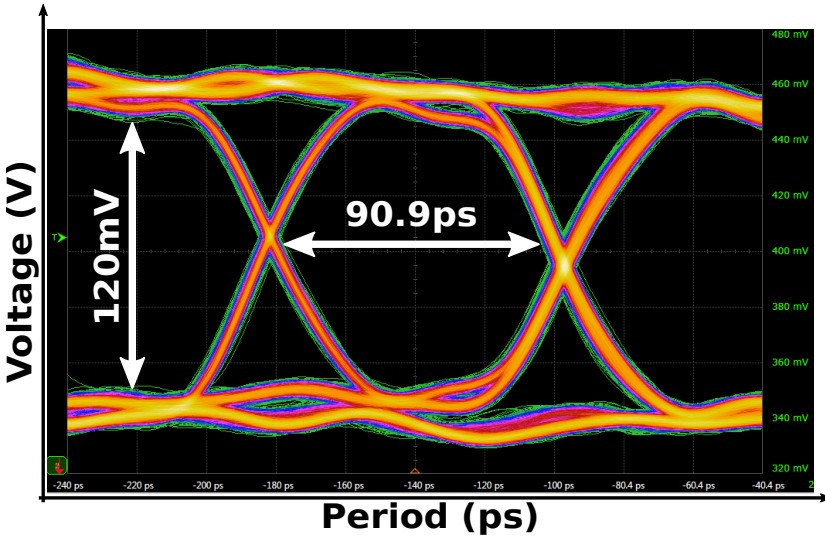

**Figure 11.** Eye diagram of the comparator output at 11 GHz for a coherent Nyquist input frequency.

This work compares favorably with state-of-the-art comparators, summarized in Table 1. This design achieved the highest reported operating frequency and the lowest delay slope, as well as the smallest sensitivity to $V_{DD}$ and $V_{CM}$ variations, compared to previously-measured published works. It also exhibited a very low input-referred noise and calibrated offset with state-of-the-art energy/comparison, demonstrating the effectiveness of the proposed calibration technique and its nearly zero overhead.

**Table 1.** Performance summary and comparison with state-of-the-art comparators.

|  | This Work | [11] | [21] | [29] | [30] | [31] | [22] |
|---|---|---|---|---|---|---|---|
| **Technology (nm)** | **28 nm** | 90 nm | 90 nm | 65 nm | 65 nm | 65 nm | 65 nm |
| **Supply (V)** | **1.0** | 1.2 | 1.2 | 1.2 | 1.0 | 1.2 | 1.2 |
| **Delay/log($\Delta V_{IN}$)** | **12 ps/dec** | 44 ps/dec | 24 ps/dec | 20 ps/dec | N.A. | N.A. | 16 ps/dec |
| **Maximum $F_{CLK}$ (GHz)** | **11.0** | 2.0 | 1.0 | 7.0 | 7.2 | 4.0 | 1.5 |
| **Input-referred noise (mV)** | **0.82** | 1.5 | 1.0 | 15.0 | 200.0 | 50.0 | 0.32 |
| **Sensitivity to $V_{DD}$ (mV)** | **+0.13/−0.14** | N.A. | N.A. | N.A. | N.A. | N.A. | N.A. |
| **Sensitivity to $V_{CM}$ (mV)** | **−0.19/+0.18** | N.A. | −0.2/+0.2 | N.A. | N.A. | N.A. | N.A. |
| **Uncalibrated offset (mV)** | **10.3** | 13.0 | 13.7 | 22.0 | N.A. | N.A. | 11.6 |
| **Calibrated offset (mV)** | **0.99** | 13.0 | 1.69 | 22.0 | N.A. | 3.0 | 0.53 |
| **Energy/comparison (fJ)** | **81** | 113 | 40 | 185 | 63 | 114 | 61 |

## 5. Conclusions

A high-speed, low-noise dynamic comparator with a dual-sided self-calibrating loop has been presented. The proposed dynamic calibration tremendously reduces the input offset by 10×, limited only by the comparator noise, without compromising its speed or significantly increasing its power. Combined with the implemented multi-stage comparator to enable better optimization between various trade-offs, the highest reported maximum frequency of 11 GHz is realized with only 0.82 mV and 0.99 mV input noise and offset, respectively, consuming only 0.89 mW from a 1 V supply. The prototype occupies a total area of only 0.00054 mm². In summary, the proposed circuit is an ideal candidate for any high speed, low noise/offset, power-/area-efficient mixed-signal system and can be adapted to any comparator structure. Moreover, its fully-dynamic implementation ensures 100% drawback-free scalability to lower technology nodes.

Future research will involve realizing a faster and lower noise comparator circuit to incorporate the proposed calibration loop. Finally, more transistor stacking will be employed in the circuit of Figure 3 to realize a finer calibration step and reduce the leakage on CALP/CALN without increasing $C_{CALP}/C_{CALN}$.

**Author Contributions:** A.R. designed, simulated, taped-out, and measured the circuits described in this manuscript under the supervision of M.S. (Michiel Steyaert), and F.T. and M.S. (Maarten Strackx) contributed to the technical discussions during the design time.

**Funding:** This research was partially funded by Nokia Bell Labs, Antwerp, Belgium.

**Acknowledgments:** The authors would like to thank Nokia Bell Labs, Antwerp Belgium, for partially supporting this work.

**Conflicts of Interest:** The authors declare no conflicts of interest.

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
