# Peer review of "An 11 GHz Dual-Sided Self-Calibrating Dynamic Comparator in 28 nm CMOS"

_electronics, doi:10.3390/electronics8010013_

Round 1
Reviewer 1 Report
The authors propose a comparator with offset calibration. The performance is good. However, I have two concerns on the proposed comparator.
1. The mismatch of the differential pair of the comparator is compensated by adding two transistors (MSP and MSN) with capacitors (CCALP and CCALN). The gate voltages of MSP and MSN are stored on the capacitors. Because of the leakage, the calibration should be performed often. Each calibration requires several clock cycles. If the comparator is used in an ADC and the sampling clock is used as CAL_EN, the calibration should be performed during ADC sampling cycle. However, each calibration requires several clock cycles. This greatly reduces ADC sampling rate.
2. The offset voltage is not constant. It depends on the input voltage. The offset of the proposed comparator is calibrated only for Vcom. The offset voltage The authors should show the measured offset voltages for different input common mode voltages.
Author Response
Dear Associate Editor and Reviewer,
We would like to thank you for the in-depth comments and suggestions during this revision, which will undoubtedly result in a significant quality upgrade of the final manuscript.
The file that is now provided in the submission page is the revised manuscript (AR_388044_Revised_Manuscript_with_Changes) with all the old information stroked-through in red and the new changes highlighted in blue. Both the old and the new information is kept in the submitted revised manuscript to make the changes more clearly visible to the reviewers.
For your convenience, we are also providing the comments here analytically together with their answer.
1). The mismatch of the differential pair of the comparator is compensated by adding two transistors (MSP and MSN) with capacitors (CCALP and CCALN). The gate voltages of MSP and MSN are stored on the capacitors. Because of the leakage, the calibration should be performed often. Each calibration requires several clock cycles. If the comparator is used in an ADC and the sampling clock is used as CAL_EN, the calibration should be performed during the ADC sampling cycle. However, each calibration requires several clock cycles. This greatly reduces the ADC sampling rate.
Answer:
Thank you for pointing this out. If the comparator with its proposed calibration circuit is incorporated in an ADC, then indeed the sampling clock can be used to control the calibration. Upon starting up the ADC the calibration gradually corrects the comparator offset in multiple sampling cycles, by moving small packets of charge in each cycle until the required CALP/CALN difference is reached. The several calibration cycles do not happen in the same sampling interval but over multiple sampling cycles. Only one calibration cycle occurs per sampling with a partial offset correction every time.
Therefore, this will not have any effect on the ADC sampling rate, as long as one calibration cycle stays shorter than the required sampling period. The only thing that will be affected by shorter calibration cycles (smaller portion of the total offset corrected in each of them) will be that the ADC will probably need a larger startup time until correct samples can be processed. This is never a problem since, for high-speed ADCs (intention of this work) that startup time will never be above several seconds, which is fully transparent to the user.
2). The offset voltage is not constant. It depends on the input voltage. The offset of the proposed comparator is calibrated only for Vcom. The authors should show the measured offset voltages for different input common mode voltages.
Answer:
Thank you very much for this comment. Indeed when the common mode of the comparator increases, the offset mainly of the input transistors increases due to the AVT and Aβ change because of the different biasing condition. As long though as the increased offset remains within the maximum calibration range that is determined by the size ratio of MSP-MSN and M1P-M1N, this offset can be compensated down to the noise level. Of course, the noise of the comparator will also increase with increasing common mode, therefore the compensated offset will result in a higher value since the noise will be higher.
For this reason, the offset simulation previously shown only for a VCM of 0.5V (Fig. 4c in the initial manuscript) has now been extended for VCM values of 0.4V and 0.6V as well and the complete offset simulation is illustrated in a new figure (Fig. 5a,b,c) in the revised manuscript. It can be seen there that the offset of all the comparators increases with increasing VCM and decreases with decreasing VCM (as expected) and that the calibration loop remains functional and minimizes the offset to the noise level, which also changes with changing VCM. Therefore, the final calibrated value is changing, but the calibration loop is still operating as expected.
Unfortunately, further measurements were not possible for the limited time available for this revision, due to a high occupation of the necessary equipment in the lab by other colleagues. The authors hope that the extra provided simulations, together with the above explanation will convince the reviewer about the functionality of the proposed circuit under different common mode conditions. Common mode values that do not deviate too much from the mid-supply level have been investigated all along since such values result in balanced comparator noise/speed trade-offs as well as trade-offs of other blocks when the comparator is used in an ADC (such as T/H swing, DAC generated range etc).
We would like to express once more our gratitude for the valuable feedback which will improve the technical completeness of the manuscript.
Sincerely,
Athanasios Ramkaj
Michiel Steyaert
Filip Tavernier

Reviewer 2 Report
The proposed dual-sided self-calibrating dynamic comparator looks very interesting since it achieve good noise performances and lower power consumption with a limited area. However, I have some comments: 1) It will be easier for reader to understand better your work if you can better appeal it compared to others. In page 7, comparison table to other works form literature is very useful but I think you need to mention in the abstract too (line 5…). 2) Adding more details when explaining simulation and measurement results will be also helpful. 3) Professional proof reading is required to fix many grammar and typing errors.
Author Response
Dear Associate Editor and Reviewer,
We would like to thank you for the in-depth comments and suggestions during this revision, which will undoubtedly result in a significant quality upgrade of the final manuscript.
The file that is now provided in the submission page is the revised manuscript (AR_388044_Revised_Manuscript_with_Changes) with all the old information stroked-through in red and the new changes highlighted in blue. Both the old and the new information is kept in the submitted revised manuscript to make the changes more clearly visible to the reviewers.
For your convenience, we are also providing the comments here analytically together with their answer.
1). It will be easier for the reader to understand better your work if you can better appeal it compared to others. In page 7, comparison table to other works from literature is very useful but I think you need to mention in the abstract too (line 5…).
Answer:
Thank you very much for this suggestion. This has been taken into account and some more emphasis is put on this work’s highlights by adding more details in the abstract (page 1). This accompanies the already existing text where the comparison table is discussed.
2). Adding more details when explaining simulation and measurement results will be also helpful.
Answer:
Thank you for this comment. Taking this into account, some extra explanation is provided where the simulations are presented (Section 3.2) (more details as to how they are performed) and in the measurements’ section (Section 4).
3). Professional proof reading is required to fix many grammar and typing errors.
Answer:
Thank you very much for your comment. The revised manuscript has been proofread to a greater extent compared to the initial one, and online grammar tools have been used to correct some of the language mistakes. The changes are highlighted in the revised manuscript. We hope the grammar and syntax will be more correct now.
We would like to express once more our gratitude for the valuable feedback which will improve the technical completeness of the manuscript.
Sincerely,
Athanasios Ramkaj
Michiel Steyaert
Filip Tavernier

Round 2
Reviewer 1 Report
The revised manuscript can be published in this Journal now.
Author Response
Dear Associate Editor and Reviewer,
We would like to express once more our cordial appreciation for the comments received during this revision, which will undoubtedly result in a significant quality upgrade of the manuscript as a total.
Following your suggestion, the manuscript has gone through an online spell check and some minor mistakes have been corrected. Furthermore, based on the suggestions from the academic editors, a "Measurement Setup" subsection has been added in the final manuscript with a block diagram of the setup and some explanation of the procedure followed. Finally, two more sentences have been added to the "Conclusions" section and the reference number has been increased to the Editors' suggestions.
The file that is now provided in the submission page is the final revised manuscript (AR_388044_Final_Manuscript) with the aforementioned additions highlighted.
We would like to thank you once more for allocating some of your limited time to increase our manuscript's quality and technical completeness.
Sincerely Yours,
Athanasios Ramkaj
Michiel Steyaert
Filip Tavernier